# A Bridge Too Far? Towards Medical Therapy for Clinically Nonfunctioning Pituitary Tumors

**DOI:** 10.3390/ijms26125898

**Published:** 2025-06-19

**Authors:** Nikita Mogar, Dongyun Zhang, Anthony P. Heaney

**Affiliations:** 1UCLA Department of Medicine, Los Angeles, CA 90095, USA; nmogar@mednet.ucla.edu (N.M.); dongyunzhang@mednet.ucla.edu (D.Z.); 2UCLA Department of Neurosurgery, Los Angeles, CA 90095, USA

**Keywords:** clinically nonfunctioning pituitary tumors, medical treatment, surgery, radiation, somatostatin receptor ligand, dopamine receptor agonist, molecular pathogenesis

## Abstract

Clinically nonfunctioning pituitary tumors (CNFPTs) typically do not cause hormonal excess, progress insidiously, and are often large and invasive at presentation. Complete resection is frequently not attainable; radiotherapy (RT) may effectively limit growth but carries a significant risk of hypopituitarism. Medical therapy with dopamine D2 receptor agonists and/or somatostatin analogs has been explored in CNFPTs but have yielded inconsistent results, and there is an unmet need for novel efficacious and safe medical therapies. The authors used the PubMed database to identify and review articles published from January 1982 to July 2024, that discussed the medical treatment of CNFPTs. The most commonly studied medical therapies were somatostatin receptor ligands (SRLs) and dopamine D2 receptor agonists. Of 111 patients with CNFPTs treated with SRLs, 31 (28%) exhibited tumor shrinkage. Following dopamine agonist treatment in 355 patients, tumor shrinkage occurred in 113 (32%), tumor stabilization in 182 (51%), and tumor growth in 60 (17%). The efficacy of other less commonly employed therapies such as GnRH analogs, PRRT, and temozolomide was also reviewed. Efficacious and safe medical therapies evaluated in robust randomized placebo-controlled clinical trials are needed to improve the management of CNFPTs.

## 1. Introduction

Clinically nonfunctioning pituitary tumors (CNFPTs) account for 14–54% of all pituitary tumors, grow insidiously, and are typically found incidentally during cranial imaging or when patients present with visual field defects and/or hypopituitarism [1,2]. CNFPTs can rarely present with pituitary apoplexy (8%) or hyperprolactinemia from stalk displacement [3,4]. The complete surgical resection of CNFPTs is frequently unattainable, and ~65% of patients will harbor remnant tumor after surgery with a >40% risk of regrowth over 5–10 years [4,5,6]. Advances in histopathological classification have provided a more detailed characterization of the cellular origin of CNFPTs but molecular studies have not yet guided choice of medical therapies. Medical therapies that have primarily been used for CNFPT treatment include dopamine agonists and somatostatin receptor (SSTR) ligands.

This review highlights some of the challenges for drug discovery for CNFPTs given their heterogeneity, lack of circulating biomarkers, and assessment of tumor responses. There is a renewed interest in discovering innovative medical therapies for this comparatively common but elusive subset of pituitary tumors, and this review will hopefully spur efforts in this field to identify novel therapeutic options for patients.

## 2. Materials and Methods

The authors used the PubMed database to identify and review articles published from January, 1982, to July, 2024, using the search terms “CNFPTs,” “clinically nonfunctioning pituitary tumors,” “NFPAs,” and “nonfunctioning pituitary adenomas,” in combination with “dopamine agonist,” “somatostatin ligand,” “GnRH analog,” PRRT,” “peptide receptor radionuclide therapy,” “temozolomide,” “folate receptors,” “mTOR inhibitors,” and “VEGF inhibitors.” This literature review encompasses studies that were conducted worldwide, reflecting its global scope. The studies that were ultimately included were conducted in a variety of countries, including the United States, Portugal, Spain, Japan, and Italy. Additionally, all studies were accessed in English. Statistical or subgroup analyses were not possible, as the studies employed different methods of measurement of tumor size, medications and doses, duration of treatment, and definitions of significant change in tumor size.

## 3. Results

### 3.1. CNFPTs Represent a Heterogenous Group of Pituitary Tumors

The availability of immunohistochemical evaluation for pituitary-derived hormones and cell lineage-specific transcription factors in resected pituitary tumors has demonstrated that CNFPTs comprise a range of pituitary tumor subtypes (Table 1) [7,8]. This recognition that they are not a single entity may have implications for the development of medical therapy unless a common therapeutic target is identified.

#### 3.1.1. Gonadotroph Tumors

Gonadotroph tumors are the most common subtype of CNFPTs and the second most common pituitary tumor subtype after lactotroph tumors [9]. They are immunoreactive to the transcription factor steroidogenic factor 1 (SF-1/NR5A1) and/or the gonadotropins follicular stimulating hormone (FSH) and luteinizing hormone (LH) [10]. The vast majority of gonadotroph tumors (~90%) are not associated with a clinical syndrome; but rarely cause ovarian hyperstimulation in women and sexual dysfunction in men due to excessive gonadotropins [11]

#### 3.1.2. Silent Corticotroph Tumors

Silent corticotroph tumors account for ~30% of all corticotroph tumors and 10–20% of silent pituitary tumors and originate from a TBX19/Tpit expressing cell lineage [12]. Some cause elevated circulating adrenocorticotropic hormone (ACTH) levels, though manifestations of hypercortisolism are infrequently seen due to the incomplete processing of the proopiomelanocortin (POMC) peptide [12]. The rare Crooke’s cell adenoma subtype of silent corticotroph tumors exhibits higher proliferative and recurrence rates [11].

#### 3.1.3. Silent Somatotroph Tumors

Silent somatotroph tumors, immunoreactive to POU class 1 homeobox 1 (POU1F1/Pit-1) and growth hormone (GH) at variable levels, are exceptionally rare and account for 2–4% of all types of pituitary tumors [13]. They typically do not cause increased GH and insulin like growth factor-1 (IGF-1) levels or cause acromegalic features [14]. They are often resistant to somatostatin analog treatment due to low somatostatin receptor 2 (SSTR2) expression [15].

#### 3.1.4. Silent Thyrotroph Tumors

Functioning thyrotroph tumors are very rare, accounting for 1 to 2% of all pituitary adenomas, and silent thyrotroph tumors are even less common [16]. These thyrotroph tumors express either the thyroid stimulating hormone (TSH) α- and β-subunits and/or POU1F1/Pit-1 and/or GATA binding protein 2 (GATA2). They typically produce dysfunctional TSH and do not cause hyperthyroxinemia [16].

#### 3.1.5. Silent Lactotroph Tumors

Rarely, silent lactrotroph tumors that express POU1F1/Pit-1 and estrogen receptor alpha (ERα) but not prolactin (PRL) are encountered and can cause a mild elevation of prolactin levels due to stalk displacement [17]. They can occasionally be found as an admixture with GH-expressing cells in mammosomatotroph tumors and Pit-1 + plurihormonal tumors [18,19].

#### 3.1.6. Null Cell Tumors

True null cell tumors account for 1–5% of CNFPTs and express the neuroendocrine marker synaptophysin (SYP), but not any pituitary lineage specific transcription factors (POU1F1, SF1, TBX19, and ERα) and/or hormones (GH, PRL, TSH, FSH-β, LH-β, ACTH, and α-subunit) [20,21].

### 3.2. Current Management of CNFPTs

Therapy for CNFPTs should include consideration of tumor size, the effect on the optic tract, pituitary gland, or cranial nerves, patient co-morbidities, and patient wishes. In asymptomatic and small CNFPTs, it may be appropriate to monitor endocrine function and radiologic tumor size annually without intervention [22]. Asymptomatic patients with macroadenomas without visual threat or significant comorbidities may also warrant a conservative approach, though imaging may be more frequent (9–12 months) along with the monitoring of visual fields [23]. Fortunately, pituitary surgery in expert centers is a comparatively safe procedure with low complication (<5%) and mortality rates (<1%) [24]. Thus, surgical resection is generally offered to younger patients even with asymptomatic CNFPT macroadenomas as the tumors have a longer time to grow [23]. Surgery is indicated in any patient in whom compressive symptoms on the optic chiasm or cranial nerves can be alleviated by tumor debulking [23]. The role of surgery in reversing established hypopituitarism in these patients is not clear, with one meta-analysis reporting only one third of patients with an improvement in pituitary function following tumor resection, with a 4.9- and 2.5-fold risk of developing new anterior pituitary dysfunction and permanent central arginine vasopressin (AVP)-deficiency, respectively, following the surgery [24].

Surgical gross total resection (GTR) or near-total resection of CNFPTs ranges from 20 to 80% [25,26] but growth of the remnant tumor is seen in 30–50% of patients over five to ten years [4,5,6,25]. Currently, options for recurrence include further surgical debulking or adjunctive radiotherapy (RT). RT effectively stops further tumor growth in 85% to 95% of patients at five to ten years [27]. However, radiation carries a risk of hypopituitarism in up to 40% of patients, though this risk may be less in patients where the radio-target is distant from the pituitary stalk and the normal pituitary gland [28,29].

### 3.3. Current Role of Medical Therapy in CNFPTs

While the use of immunohistochemical transcription marker analysis and pituitary hormone assessment in CNFPTs has improved our ability to sub-categorize these tumors, there are no circulating or tissue biomarkers that can be monitored in parallel with tumor growth. Additionally, the presence of several distinct CNFPT subtypes may make finding a single efficacious therapy for all elusive. Nonetheless, given the frequent inability to achieve total tumor resection in CNFPTs, their regrowth rates, and risk of hypopituitarism and secondary brain tumors after adjuvant radiation, there is a clear unmet need for medical therapies for these common pituitary tumors. Medical therapy for CNFPTs has thus far been directed at the somatostatin receptors (primarily subtype 2 [SSTR-2]) and dopamine receptors (subtype 2 [DR2)]), with some use of gonadotrophin-releasing hormone (GnRH) analogs.

#### 3.3.1. Somatostatin Receptor Ligands

Somatostatin binds SSTRs to induce cell cycle arrest (SSTR-1, -2, -4, and -5) and apoptosis (SSTR-3) and inhibit vascular endothelial growth factor (VEGF) and α-subunit secretion [30,31]. Studies have reported the expression of SSTR subtypes in CNFPTs, predominantly SSTR-2 and SSTR-3, with a minority reporting SSTR-5 [11,32,33,34,35]. Somatostatin receptor ligand (SRL) therapy with octreotide and lanreotide, (targeting SSTR-2 and -5) and pasireotide (targeting SSTR-1, -3, and -5) has been evaluated in CNFPTs [5,36,37,38]. Table 2 summarizes the results of eight studies from the literature that evaluated the efficacy of somatostatin receptor ligands in treating CNFPTs. In total, 31/111 patients (28%) with CNFPTs treated with SRLs exhibited tumor shrinkage. The number of patients who had tumor stabilization or tumor growth was not possible to calculate as some studies did not carefully distinguish between tumor stabilization and tumor growth.

One review of 11 studies highlighted the challenges in comparing different parameters (diameter, volume, and/or area) to assess tumor size. Overall, this analysis reported that only 12% of cases treated with octreotide exhibited a reduction in tumor size, with the majority exhibiting stable tumor sizes [36]. In another study of eight patients with CNFPTs treated with octreotide 100 mcg TID for 3–6 months, 3/8 patients had tumor diameter reductions (median reduction of 5 mm), 1/8 exhibited a stable tumor, and 4/8 exhibited tumor growth (median increase of 1.5 mm, interquartile range (IQR) [1, 2]) [39]. In one study of 19 patients with CNFPTs treated with octreotide 150–300 mcg daily for 1–12 months, 15/19 patients exhibited a decreased tumor area (median reduction of 12 mm^2^, IQR [−6–−30]), 0/19 exhibited stable tumors, and 4/19 exhibited tumor growth (median growth of 5 mm^2^, IQR [2–14]). However, no statistically significant change in the average cross-sectional area was found [40]. In one study of nine patients with CNFPTs treated with octreotide 0.05 mg three times weekly to 0.6 mg daily, 3/9 had reductions in tumor size (maximum reduction of 30%), though the number of patients with stable tumors or growth was not documented [41]. A later review paper from the same authors reported two patients with tumor shrinkage, seven patients with stable tumors, and no patients with tumor growth [36]. In another study of 14 patients treated with octreotide at 300–1500 mcg per day for three months, 2/14 patients had tumor shrinkage (maximum reduction of 96%, though the treatment dosage and duration were not carefully documented) [42] and the number of patients with tumor stabilization or growth was not documented. In a further study of seven patients with CNFPTs treated with octreotide 100 mcg daily to 200 mcg three times daily for up to two months, 3/7 patients exhibited tumor shrinkage by 26–73% (median change, −38%), 3/7 had stable tumors, and 1/7 had tumor growth, by 48% [43]. The GALANT double-blind, placebo-controlled, phase 3 clinical trial compared the efficacy of 72 weeks of treatment with lanreotide 120 mg (22 patients [50%]) versus placebo (22 patients [50%]) administered every 28 days on tumor diameter and volume in 44 patients with DOTATATE-PET-positive residual CNFPTs [44]. The study found no clinically significant change (defined as an increase or decrease of ≥2 mm in diameter or ≥20% in volume) in tumor size in patients treated with lanreotide (mean [SD] change + 1.2 [2.5] mm) versus placebo (+1.3 [1.5] mm) [45]. However, 3/22 (14%), 8/22 (36%), and 11/22 (50%) of patients treated with lanreotide exhibited significant tumor shrinkage, stabilization, and growth, respectively, compared to 0/22, 13/22 (64%), and 8/22 [36%] in placebo-treated patients. The authors concluded that they could not recommend SSA therapy in the treatment of residual CNFPTs.

**Table 2 ijms-26-05898-t002:** Somatostatin receptor ligand (SRL) efficacy on size of clinically nonfunctional pituitary tumors (CNFTs).

First Author (Year) Reference	Number of Treated Patients with CNFTs	Medication	Dose	Duration of Therapy (Mths)	Number of Patients (%), [Median Change (IQR)] Exhibiting Tumor Shrinkage; Measurement Method	Number of Patients (%) Exhibiting Stable Tumor Size	Number of Patients (%), [Median Change (IQR)] Exhibiting Tumor Growth; Measurement Method
Katznelson (1992) [31]	6	Octreotide SC	50 µg BID–250 µg TID	2	2, [insuff]; V (%)	4	0, [NA]; V (%)
Gasperi (1993) [39]	8	Octreotide SC	100 µg TID	3–6	3, [−5 (insuff)]; D (mm)	1	4, [1.5 (1–2)]; D (mm)
Merola (1993) [40]	19	Octreotide SC	150–300 µg QD	1–12	15, [−12 (−6–−30)]; A (mm^2^)	0	4, [5 (2–14)]; A (mm^2^)
Plockinger (1994) [42]	14	Octreotide SC	300–1500 µg QD	3	2, [−76.5 (insuff)]; V (%)	12	0, [NA]; V (%)
Warnet (1997) [43]	7 with imaging before and after treatment	Octreotide SC	100 µg QD–200 µg TID	2	3, [−38 (insuff)]; V (%)	1	3, [48 (insuff)]; V (%)
Colao (1999) [41]	9	Octreotide SC	50 µg TID–600 µg QD	6–12	3, [insuff]; V (%)	insuff	insuff, [insuff]; V (%)
Fusco (2012) [5]	26	Octreotide LAR	20 mg Q1M	37 ± 18	0, [NA]; D (mm)	21	5, [insuff]; D (mm)
Boertien (2024) [45]	Lanreotide (n = 22) Placebo (n = 22)	Lanreotide LAR SC	120 mg Q28D	18	Lanreotide 3/22 (14%), [insuff]; D (mm), V (mm^3^) Placebo 0/22 (0%), [NA]; D (mm), V (mm^3^)	Lanreotide 8/22 (36%), [insuff]; D (mm), V (mm^3^) Placebo 14/22 (64%), [insuff]; D (mm), V (mm^3^)	Lanreotide 11/22 (50%), [insuff]; D (mm), V (mm^3^) Placebo 8/22 (36%), [insuff]; D (mm), V (mm^3^)
Total Number (%)	111				31 (28%)	insuff	insuff

Legend: SSTR, somatostatin receptor; LAR, long-acting release; mg, milligrams; µg, micrograms; mm, millimeters; BID, twice a day; TID, thrice a day; QD, daily; Q1M, monthly; Mths, months; V, volume; D, dimension; A, area; NA, not applicable; IQR, interquartile range; insuff, insufficient data.

The Passion 1 (NCT01283542) open-label single arm study is assessing the efficacy and safety of pasireotide LAR 60 mg every 28 days for 24 weeks on tumor volume and hormone levels in 20 patients with CNFPTs [46].

#### 3.3.2. Dopamine Agonist Therapy

Dopamine receptors include five subtypes, D1-5. D2 receptor agonists are highly effective in controlling prolactin secretion and tumor size in prolactinomas [47,48], and beneficial actions of D2 agonists have also been reported in GH- and ACTH-secreting pituitary tumors [49,50]. It is reported that 67% of CNFPTs express the D2 receptor.

One study summarized 27 studies between 1979 and 2016 of dopamine agonist therapy (bromocriptine, quinagolide, and/or cabergoline) for CNFPTs, reporting that 41/122 (33.6%) patients exhibited tumor shrinkage, 57/122 (46.72%) exhibited tumor stabilization, and 24/122 (19.67%) experienced tumor growth [51]. Another review of 16 articles by Cooper et al. found that CNFPTs in 30% of patients treated with dopamine agonists exhibited tumor shrinkage, 58% remained stable, and 12% exhibited tumor growth despite treatment [52].

Table 3 summarizes the results of 18 studies including 355 patients with CNFPTs treated with dopamine agonists. A total of 113/355 (32%) patients exhibited tumor shrinkage, 182/355 (51%) exhibited tumor stabilization, and 60/355 (17%) exhibited tumor growth [50].

Older studies examining the effect of bromocriptine on CNFPT size reported between 0 and 81% of patients achieved tumor reduction [53]. Unfortunately, four out of six studies did not report the exact change in tumor area, volume, or diameter [53,54,55,56,57,58]. Two additional studies found that quinagolide induced tumor shrinkage in 20–33% of 16 patients (Table 3) [59,60].

In a study of 13 patients treated with cabergoline 0.25 mg per week, 7/13 had tumor shrinkage (maximum reduction of 18%, median change of −15%, IQR [−12–−18]) [61], 5/13 had stable tumors, and 1/13 had tumor growth (median increase 25%), respectively. One study of 33 patients with visible post-surgical residual CNFPTs divided patients into a “primary prevention” group without documented tumor growth (Group I) and a secondary prevention group who had exhibited tumor growth (Group II). These patients were compared to two control groups including patients with documented tumor growth who declined treatment (Control for Group II) and matched patients who had residual tumor but had not exhibited tumor growth (Control for Group I). In the preventative group, tumor diameters reduced in 9/20 patients (45%, compared to 0% in both control groups), remained stable in 9/20 (45%), and grew in 2/20 (10%). In the secondary prevention group, tumors shrank in 15%, remained stable in 46%, and grew in 39% of patients [62]. This study suggested that early treatment with dopamine agonists may help to mitigate the growth of CNFPTs.

A further study measuring tumor diameter and volume in nine patients with post-surgical residual CNFPTs treated with cabergoline (1 to 3 mg/week) reported that 56% (median change −49.3%, IQR [−38.9–−57.2]), 0%, and 44% of the patients exhibited tumor shrinkage, tumor stabilization, and tumor growth (median change 8.7%, IQR [6.2–29.5]), respectively [50].

In a second larger study of 79 patients with ACTH- and GH-negative CNFPTs, 55 patients (preventative group) were treated after surgery with non-standardized doses of dopamine agonist therapy for residual tumors). In total, 24 patients (the remedial group) were treated with D2 agonists after initially declining therapy. No tumor growth was seen in 30% of the remedial group, 87% of the preventative group, and 47% of the untreated control group [60]. Tumor shrinkage was seen in 7 (29.2%), 21 (38.2%), and 0 (0%) of patients in the remedial, preventative, and control groups, respectively. The authors concluded that early adjunctive medical therapy with D2 agonist therapy reduced the need for additional surgery and/or radiotherapy from 47 to 16%, and tumor response did not appear to be associated with DR2 expression.

An additional randomized open-label two-year clinical trial compared cabergoline (n = 59) to no intervention (n = 57) in 116 patients with residual CNFPTs after transsphenoidal surgery [36,63]. Overall, 28.8%, 66.1%, and 5.1% of patients in the CBG-treated group demonstrated tumor shrinkage ≥ 25%, stabilization < 25%, and enlargement ≥ 25%, respectively, in comparison to (10.5% shrinkage, 74% stability, and 16% growth in the control group [*p* = 0.01]). Baseline patient characteristics were similar and tumor response did not appear to correlate with DR2 expression [36,59,64].

**Table 3 ijms-26-05898-t003:** Dopamine agonist (DA) efficacy on size of clinically nonfunctional pituitary tumors (CNFTs).

First Author (Year) Reference	Number of Treated Patients with CNFTs	Medication	Dose	Duration of Therapy (Mths)	Number of Patients (%), [Median Change (IQR)] Exhibiting Tumor Shrinkage; Measurement Method	Number of Patients (%) Exhibiting Stable Tumor Size	Number of Patients (%), [Median Change (IQR)] Exhibiting Tumor Growth; Measurement Method
Wollesen (1982) [53]	11	BCP	15–60 mg/d	2–33	9, [−32 (−21.5–−41.5)]; A (%)	0	2, [14 (insuff)]; A (%)
Barrow (1984) [54]	7	BCP	2.5–7.5 mg/d	1.5	1, [−2.1 (insuff)]; A (cm^2^)	5	1, [0.1 (insuff)]; A (cm^2^)
Pullan (1985) [55]	5	BCP	15–37.5 mg/d	3–27	1, [insuff]; NA	4	0 [NA]; NA
Verde (1985) [56]	20	BCP	7.5–20 mg/d	1–32	1, [insuff]; NA	19	0 [NA]; NA
Zarate (1985) [57]	7	BCP	15–22.5 mg/d	0.5–12	0, [NA]; NA	7	0 [NA]; NA
van Schaarde-nburg (1989) [58]	25 (11/25 had imaging)	BCP	5–22.5 mg/d	1–73	2, [insuff]; NA	7	2 [insuff]; NA
Ferone (1998) [60]	6	QUI	300–600 µg/d	6–12	2, [insuff]; NA	4	0 [NA]; NA
Colao (2000) [65]	10	QUI CBG	600 µg/d (QUI) 3 mg/w (CBG)	12	2, [insuff]; V (%)	8	0 [NA]; V (%)
Nobels (2000) [59]	10	QUI	300 µg/d	36–93	2, [−1.8 (insuff)]; V (cm^3^) [−37.3% (insuff)]; V (%)	0	8, [2.3 cm^3^ (1.1–3.5)]; V (cm^3^) [31.3% (13.1–54.8)]; V (%)
Lohmann (2001) [61]	13	CBG	0.25–1.0 mg/w	12	7, [−15 (−12–−18)]; V (%)	5	1, [25 (insuff)]; V (%)
Pivonello (2004) [50]	9	CBG	1–3 mg/w	12	5, [−49.3 (−38.9–−57.2)]; V (%)	0	4, [8.7 (6.2–29.5)]; V (%)
Greenman (2005) [62]	Preventative (I): n = 20 Secondary (II): n = 13 Control I (CI): n = 47 Control II (CII): n = 38	BCP QUI CBG	10 mg QD (BCP) 300 mg QD (QUI) 1.5 mg/week (CBG)	40 ± 48	I: 9/20 (45%), [insuff]; D (mm) II: 2/13 (15%), [insuff]; D (mm) CI: 0 (0%), [NA]; D (mm) CII: 0 (0%), [NA]; D (mm)	I: 9/20 (45%), [insuff]; D (mm) II: 6/13 (46%), [insuff]; D (mm) CI: 18/47 (38%), [insuff]; D (mm) CII: 18/38 (47%), [insuff]; D (mm)	I: 2/20 (10%), [insuff]; D (mm) II: 5/13 (39%), [insuff]; D (mm) CI: 29/47 (62%), [insuff]; D (mm) CII: 20/38 (53%), [insuff]; D (mm)
Garcia (2013) [64]	19	CBG	2 mg/w	6	9, [−26.7 (−16.7–−43.4); V (%)	6	4, [14.8 (12.5–74.4)]; V (%)
Vieira Neto (2015) [66]	9	CBG	3 mg/w	6	8, [−29.2 (−18.9–−39.5)]; V (%) by 3D [−17.3 (−9.6–−30.9)]; V (%) by Di Chiro and Nelson	0	1, [2.6 (insuff)]; V (%) by 3D [0.2 (insuff)]; V (%) by Di Chiro and Nelson
Greenman (2016) [6]	Preventative (P) (n = 55) Remedial (R) (n = 24) Control (C) (n = 60)	CBG BCP	10 mg QD (BCP) 2 mg/w (CBG)	105.6 ± 78	P: 21/55 (38%), [insuff]; D (mm) R: 7/24 (29%), [insuff]; D (mm) C: 0/60 (0%), [NA]; D (mm)	P: 27/55 (49%), [insuff]; D (mm) R: 7/24 (29%), [insuff]; D (mm) C: 28/60 (47%), [insuff]; D (mm)	P: 7/55 (13%), [insuff]; D (mm) R: 10/24 (42%), [insuff]; D (mm) C: 32/60 (53%), [insuff]; D (mm)
Batista (2019) [63]	Medical Therapy (MT) (n = 59) Control (C) (n = 57)	CBG	0.5 mg/d	24	MT: 17/59 (29%), [insuff]; V (%) C: 6/57 (11%), [insuff]; V (%)	MT: 39/59 (66%), [insuff]; V (%) C: 42/57 (74%), [insuff]; V (%)	MT: 3/59 (5%), [insuff]; V (%) C: 9/57 (16%), [insuff]; V (%)
Iglesias (2022) [67]	Medical Therapy (CAB) (n = 22) Observation (OBS) (n = 40)	CBG	0.5–1.5 mg/w	13 (10.5–17)	CAB: 3/22 (14%), [insuff]; D (mm) OBS: 2/40 (5%), [insuff]; D (mm)	CAB: 17/22 (77%), [insuff]; D (mm) OBS: 32/40 (80%), [insuff]; D (mm)	CAB: 2/22 (9%), [insuff]; D (mm) OBS: 6/40 (15%), [insuff]; D (mm)
Ayalon-Dangur (2024) [68]	25	CBG	≥ 1 mg/w	24	5, [6 (−2–−75)]; D (mm)	12	8, [4 (2.5–8)]; D (mm)
Total Number (%)	355				113 (33%)	182 (51%)	60 (17%)

Legend: QUI, quinagolide; CBG, cabergoline; BCP, bromocriptine; mg, milligrams; µg, micrograms; QD, daily; d, day; w, week; mths, months; V, volume; D, dimension; A, area; NA, not applicable; 3D, 3-dimensional; IQR, interquartile range; insuff, insufficient data.

One study of 36 patients with CNFPTs (9/36 patients received post-operative cabergoline 3 mg/week for at least 6 months), 8/9 (88.9%) patients exhibited a tumor volume reduction using two methods (median change −29.2%, IQR [−18.9–−39.5] by three-dimensional [3D] measurements; and median change −17.3%, IQR [−9.6–−30.9] by Di Chiro and Nelson measurements), 0 patients exhibited tumor stabilization, and 1 patient exhibited tumor growth (2.6% by 3D and 0.2% by Di Chiro and Nelson). Significant tumor volume reduction was arbitrarily defined as a reduction in tumor volume ≥25%, which was observed in six of nine (67%) patients [66].

In another study where 22/62 patients with residual CNFPTs (35.5%) were treated with cabergoline (compared with a placebo in 40/62 (64.5%)) [67], 3/22 (13.6%), 17/22 (77.2%), and 2/22 (9.1%) patients experienced tumor shrinkage, stabilization, and growth, respectively, compared to 2/40 (5%), 32/40 (80%), and 6/40 (15%) in the placebo group [4].

A recent study examined 25 surgery-naive patients with CNFPTs ≥1 cm treated with cabergoline ≥1 mg per week for at least 24 months (range 1–2.5 mg/week, [median 1.5 mg/week]). Change in tumor size was defined as ≥2 mm. In total, 5/25 patients (20%), 12/25 (48%), and 8/25 (32%) exhibited tumor size reduction (median change −6 mm, IQR [−2–−7.5]), tumor stabilization, and tumor growth (median increase 4 mm, IQR [2.5–8]), respectively. A total of 6/7 patients (28%) had elevated prolactin levels (1.5–2.5-times the upper limit of normal) and a further patient had a four-fold elevation in the serum prolactin level [68].

Some studies have examined combination therapy of SRLs with D2 agonists. In one study of 10 patients with CNFPTs treated with octreotide 200 mcg TID and cabergoline 0.5 mg daily, 6/10 patients exhibited a mean tumor shrinkage of 30% ± 4% with a range of 18–46%, tumor stabilization was seen in 3/10, and tumor growth in 1/10 patients [69]. There is an ongoing phase 2 open-label clinical trial (NCT01620138) that will compare tumor responses to combination therapy with cabergoline (3 mg/week) and pasireotide (900 mcg BID) for 6 months in 21 patients with CNFPTs.

#### 3.3.3. GnRH Analog Therapy

As discussed above, the majority of CNFPTs express gonadotropins or their α- or β-subunits, and so GnRH analogs have also been explored as a treatment for CNFPTs. Data on the effect of GnRH analogs on CNFPTs is not well described in the literature, and the existing studies point to an equivocal effect on circulating α-subunit or gonadotrophin levels [30,36,70,71,72,73,74,75,76]. Changes in tumor size were not carefully documented.

#### 3.3.4. Peptide Receptor Radionuclide Therapy and Temozolomide

Peptide receptor radionuclide therapy (PRRT) involves linking the radionuclide lutetium-177 to a targeting somatostatin analog, and it is effective in treating a variety of neuroendocrine tumors [77]. PRRT using lutetium-177 linked to octreotide has been used in six “aggressive” CNFPTs, where three out of six PRRT-treated patients with CNFPTs exhibited stable disease, two had progressive disease, and one patient died [78].

Finally, the alkylating chemotherapeutic agent temozolomide has been used in the treatment of pituitary carcinomas and “aggressive” pituitary tumors of various subtypes including CNFPTs. There is some suggestion that CNFPTs may be less responsive to temozolomide based on one study in 110 patients where only 17% of patients with CNFPTs demonstrated tumor regression compared to 45% of patients with functional pituitary tumors [79]. A second study summarizing eight prior case series of 100 patients with pituitary tumors treated with temozolomide found that of 27 patients with CNFPTs, 7 (22%) demonstrated tumor regression, 13 (48%) had stable disease, and 8 (30%) had tumor growth [80]. One retrospective study of 28 patients with aggressive pituitary tumors treated with temozolomide after surgery, 15 of whom had CNFPTs [81], found that 2/15 (13.3%), 9/15 (60%), and 3/15 (20%) patients exhibited a partial response, tumor stabilization, and tumor growth, respectively. Insufficient data was available in the final patient. In total, 4/15 patients received medical therapy with cabergoline during the course of their treatment.

### 3.4. Emerging Experimental Novel Treatment Modalities of CNFPT

#### 3.4.1. Immune Checkpoint Inhibitor Therapy

The use of immune checkpoint inhibitors (ICIs) targeting programmed death-ligand (PD-L1) and others has proven highly effective in treating several cancers including lung cancer and melanoma. To date, limited studies have examined the use of ICIs such as PD-1/PD-L1 inhibitors in pituitary neuroendocrine tumors and have been mostly limited to refractory pituitary tumors. A review of 29 studies of patients with pituitary neuroendocrine tumors treated with a variety of ICIs included four patients with CNFPTs, three of whom had metastatic disease. Two of these four patients (50%) responded partially with radiographic improvement in response to therapy. The other two patients (50%) exhibited stable disease without improvement, and none exhibited disease progression [82]. Further studies with larger sample sizes are needed to clarify the efficacy of immunotherapy in CNFPTs.

#### 3.4.2. Folate Receptor-Mediated Drug Targeting

Folate receptors (FR α, β, and γ) are cysteine-rich glycosyl-phosphatidyl-inositol-anchored membrane proteins that bind folic acid and folate conjugates, which can then facilitate de novo nucleotide synthesis, DNA-methylation, and DNA-damage repair [83]. When cognate folic acid binds the FR, the receptor–ligand complex clusters into lipid rafts, initiating a clathrin-independent, caveolae-mediated endocytosis process [84]. The plasma membrane then invaginates to form vesicles (endosomes) containing the FR–folate complex [85]. As the endosome matures, the acidic environment causes folate to dissociate from the receptor, which subsequently is recycled back to the cell surface for another round of uptake [85]. This highly efficient and selective process is frequently exploited in cancer therapy and imaging [85]. FR expression is typically restricted to the placenta, lung, kidneys, and choroid plexus, but is overexpressed in many malignancies [86]. Increased FRα mRNA and protein expression has been reported in CNFPTs compared to hormonally active pituitary tumors [87]. FRα overexpression in murine αT3 gonadotroph tumor cells in vitro increased S-phase cells and cell proliferation, in part by activation of the NOTCH3 pathway [88]. In vivo imaging using a 99 mTc-Folate SPECT/CT demonstrated high FR expression in patients with CNFPTs [89]. FRα-targeted liposomal doxorubicin treatment was also shown to inhibit proliferation and invasion in primary cultures of human CNFPTs [90,91,92], suggesting that FR-mediated tumor targeting could be a feasible novel treatment option for CNFPTs and warrant further investigation (Figure 1) [93].

#### 3.4.3. mTOR Inhibitors

Activation of the AKT/mTOR pathway in human CNFPTs has been identified by transcriptomic and proteomic profiling approaches (Figure 1) [94,95]. Treatment with the mTOR inhibitors everolimus and rapamycin inhibited in vitro proliferation in 35–70% of primary cultures of CNFPTs when used as a monotherapy [96,97], and co-treatment with somatostatin analogs (octreotide and SOM230) appeared to potentiate these mTOR anti-proliferative effects [97,98]. Up to now, isolated case reports have described the use of the mTOR inhibitor everolimus in treating refractory corticotrophs [99,100,101,102], but the efficacy of mTOR inhibition in CNFPTs is still undetermined [103].

#### 3.4.4. Targeting Tumor Angiogenesis

The normal anterior and posterior lobes of the pituitary gland are highly vascular and supplied by the superior and inferior hypophyseal arteries, respectively [103]. The portal hypophyseal veins transport endocrine hormones from the pituitary into the peripheral circulation [103]. The microvascular density of pituitary carcinomas is reportedly higher than in the normal pituitary gland, suggesting that inhibiting angiogenesis could be a way to treat some pituitary tumors [104,105]. Bevacizumab, a vascular endothelial-derived growth factor (VEGF) inhibitor, apatinib, a VEGFR2 inhibitor, and sunitinib, an oral tyrosine kinase inhibitor, are all anti-angiogenic agents that have been used to treat pituitary tumors (Figure 1) [106]. In one study of three patients with CNFPTs previously treated with temozolomide, the progression-free survival following bevacizumab ranged from 9 to 26 months [107].

## 4. Discussion

We describe the natural history of these tumors, summarize currently available therapeutic options and their efficacy, and discuss how the diversity of CNFPT subtypes potentially complicates the discovery of new effective medical therapies. The interpretation and statistical analyses of many of these studies are challenging due to varying definitions of therapeutic response, differing treatment doses and durations, variable methods of tumor size measurement, and missing data points, all of which contribute to inconsistent findings from these mostly open-label studies.

## 5. Conclusions

We conclude that robust, blinded, randomized placebo-controlled clinical trials to study the efficacy and safety of SRLs and D2 agonists in CNFPTs are needed with parallel efforts to identify novel agents and potentially study some of the additional targets discussed, such as inhibitors of angiogenic pathways, folate uptake, and/or the mTOR pathway.

## Figures and Tables

**Figure 1 ijms-26-05898-f001:**
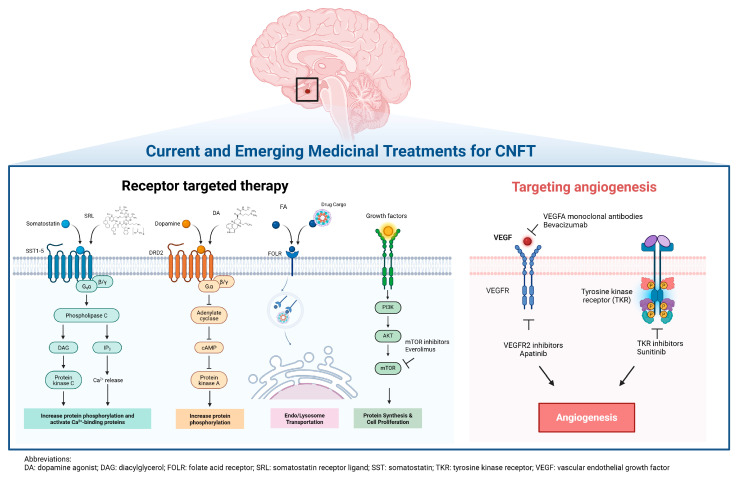
The summary of current and emerging medicinal treatment for CNFT. Created in BioRender. Zhang, D. (2025) https://BioRender.com/6wpp54q (accessed on 12 June 2025).

**Table 1 ijms-26-05898-t001:** Histological classification of clinically nonfunctioning pituitary tumors.

Tumor Type	Subtype	Frequency (% of All CNFTs)	Lineage Specific Transcription Factor	Hormone Produced
Silent Gonadotroph Tumor		40–79%	SF-1 (NR5A1), GATA2, ERa	FSHb, LHb, a-subunit
Silent Corticotroph Tumor	Sparsely Granulated Corticotroph Tumor	3–6%	Tpit (TBX19)	ACTH
	Crooke’s Cell Adenoma		Tpit (TBX19)	ACTH
Silent Somatotroph Tumor	Sparsely Granulated Somatotroph Tumor	2–4%	Pit-1 (POU1F1)	GH (or PRL)
	Mammosomatotroph Tumor		Pit-1 (POU1F1), ERa	GH and PRL (same cells)
	Mixed GH-PRL Tumor		Pit-1 (POU1F1), ERa	GH and PRL (different cells)
	Plurihormonal Tumor		Pit-1 (POU1F1), ERa	GH, PRL, TSHb
Silent Lactotroph Tumor	Sparsely Granulated Lactotroph Tumor	1%	Pit-1 (POU1F1), ERa	PRL
	Mammosomatotroph Tumor		Pit-1 (POU1F1), ERa	GH and PRL (same cells)
	Mixed GH-PRLTumor		Pit-1 (POU1F1), ERa	GH and PRL (different cells)
	Plurihormonal Tumor		Pit-1 (POU1F1), ERa	GH, PRL, TSHb
	Acidophilic Stem Cell Adenoma		Pit-1 (POU1F1), ERa	GH, PRL
Silent Thyrotroph Tumor		1–2%	Pit-1 (POU1F1), GATA2	TSHb
Null Cell Tumor		17%	None	None

## Data Availability

No new data were created in the publication of this review.

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
