# Peer review of "A Bridge Too Far? Towards Medical Therapy for Clinically Nonfunctioning Pituitary Tumors"

_ijms, 2025, doi:10.3390/ijms26125898_

Round 1

Reviewer 1 Report

Comments and Suggestions for Authors

This is an interesting and much-needed review of clinically non-functioning pituitary tumours and the current therapeutic options. It is comprehensive and well written, I have only a few minor comments on points that may add to the paper if the authors feel they have value.

  1. For SSTR ligands, the authors state that tumour stabilization or tumour growth could not be assessed as some studeis did not carefully distinguish between this- but this is assessed for dopamine agonist therapy. I am unclear why it would be possible for one and not the other, expecially as many papers cited in Tables 2 and 3 are the same? Could the authors explain this briefly- especially if this would be something that should be focused on in future studies/trials.
  2. I think in a timely and thorough review of this type, it is helpful if the authors could recommend where future studies should focus beyond the need for improved clinical trials. Where is the potential for novel therapies and/or how could these be identified through modern methods for tumour profiling (sequencing for example)? A short section on research and therapeutic perspectives might be helpful.

Author Response

Referee #1:  

Comment 1: For SSTR ligands, the authors state that tumour stabilization or tumour growth could not be assessed as some studies did not carefully distinguish between this- but this is assessed for dopamine agonist therapy. I am unclear why it would be possible for one and not the other, especially as many papers cited in Tables 2 and 3 are the same? Could the authors explain this briefly- especially if this would be something that should be focused on in future studies/trials.  

Response 1: We apologize for the oversight but Table 3 did not successfully upload as part of the prior manuscript submission. This has now been corrected and Table 3 has been uploaded with our revised manuscript.   â€¯ 

Comment 2: I think in a timely and thorough review of this type, it is helpful if the authors could recommend where future studies should focus beyond the need for improved clinical trials. Where is the potential for novel therapies and/or how could these be identified through modern methods for tumour profiling (sequencing for example)? A short section on research and therapeutic perspectives might be helpful.  

Response 2: Evaluation for somatostatin and/or dopamine receptor expression in clinically nonfunctioning tumors has been studied but have not demonstrated a consistent correlation between DR2 and/or SSTR expression and response to dopamine agonist or somatostatin receptor ligand therapy. We discuss some novel therapies that have shown promise either in vitro (folate receptor targeted therapy) or in isolated case reports (mTOR & VEGF inhibitors). We agree that advanced molecular approaches such as single cell RNA (scRNA) sequencing warrant further study and could identify additional therapeutic targets, but this has not been extensively studied as yet.  

Reviewer 2 Report

Comments and Suggestions for Authors

The authors have reviewed medical therapy for clinically non-functioning pituitary tumors. There are several flaws in this submitted manuscript, which should be corrected to be considered for publication in IJMS.

2. Materials and Methods

The rationale for the search terms is not very clear. Why was the term "Immunotherapy" not included? This could be one of the emerging treatment options.

The rationale for selecting articles limited in English, Portuguese, Japanese, Spanish, and Italian is not clear. 

The authors could consider describing the search terms used for identifying and reviewing articles on medical therapy. The current search terms used in this project may be leading to the omission of some key articles.

3.1.3. Silent Somatotroph Tumors

Reference(s) for silent somatotroph tumors (Line 91-93) are missing. It would be Reference 16.

3.1.6. Null Cell Tumors

Reference(s) for null cell tumors (Line 112-115) are missing.

3.3.1. Somatostatin Receptor Ligands

References for SSTRs expression in CNFPTs (Line 163-165) are missing, likely due to the search terms used in the project. However, it seems to result in missing key articles. The authors could consider citing the following sentences, as they report SSTRs expression profiles in nonfunctioning pituitary tumors, including silent corticotroph tumors. 

Taboada GF, Luque RM, Bastos W, Guimarães RF, Marcondes JB, Chimelli LM, Fontes R, Mata PJ, Filho PN, Carvalho DP, Kineman RD, Gadelha MR. Quantitative analysis of somatostatin receptor subtype (SSTR1-5) gene expression levels in somatotropinomas and non-functioning pituitary adenomas. Eur J Endocrinol. 2007 Jan;156(1):65-74. doi: 10.1530/eje.1.02313. PMID: 17218727.

Nielsen S, Mellemkjaer S, Rasmussen LM, Ledet T, Olsen N, Bojsen-Møller M, Astrup J, Weeke J, Jørgensen JO. Expression of somatostatin receptors on human pituitary adenomas in vivo and ex vivo. J Endocrinol Invest. 2001 Jun;24(6):430-7. doi: 10.1007/BF03351043. PMID: 11434667.

Gabalec F, Drastikova M, Cesak T, Netuka D, Masopust V, Machac J, Marek J, Cap J, Beranek M. Dopamine 2 and somatostatin 1-5 receptors coexpression in clinically non-functioning pituitary adenomas. Physiol Res. 2015;64(3):369-77. doi: 10.33549/physiolres.932821. Epub 2014 Dec 22. PMID: 25536318.

Neto LV, Machado Ede O, Luque RM, Taboada GF, Marcondes JB, Chimelli LM, Quintella LP, Niemeyer P Jr, de Carvalho DP, Kineman RD, Gadelha MR. Expression analysis of dopamine receptor subtypes in normal human pituitaries, nonfunctioning pituitary adenomas and somatotropinomas, and the association between dopamine and somatostatin receptors with clinical response to octreotide-LAR in acromegaly. J Clin Endocrinol Metab. 2009 Jun;94(6):1931-7. doi: 10.1210/jc.2008-1826. Epub 2009 Mar 17. PMID: 19293270; PMCID: PMC2730344.

Tateno T, Kato M, Tani Y, Oyama K, Yamada S, Hirata Y. Differential expression of somatostatin and dopamine receptor subtype genes in adrenocorticotropin (ACTH)-secreting pituitary tumors and silent corticotroph adenomas. Endocr J. 2009;56(4):579-84. doi: 10.1507/endocrj.k08e-186. Epub 2009 Mar 24. PMID: 19318729.

3.3.2. Dopamine Agonist Therapy

Table 3 does not show a summary of the results of patients with CNFPTs treated with dopamine agonists. It does show Table 2. 

The authors could consider adding the following two articles to review dopamine agonist therapy.

Capatina C, Poiana C. Dopamine Agonists in the Management of Non-Functioning Pituitary Adenomas. Acta Endocrinol (Buchar). 2021 Jul-Sep;17(3):377-382. doi: 10.4183/aeb.2021.377. PMID: 35342478;

Cooper O, Greenman Y. Dopamine Agonists for Pituitary Adenomas. Front Endocrinol (Lausanne). 2018 Aug 21;9:469. doi: 10.3389/fendo.2018.00469. PMID: 30186234; 

3.4.1. Folate Receptor-Mediated Drug Targeting. 

Fig. 1 suggests the FR-mediated endocytosis. However, there is no explanation in this section. Also, functions of FR in pituitary tumorigenesis are uncelar, though the section describes FR expression and its role in the activation of the Notch3 signaling pathway, leading to cell proliferation. More details would be required. 

The font size is smaller in section 4. Discussion and 5. Conclusion compared to other sections (1. Introduction, 2. Materials and Methods, 3. Results), which is strange. 

3.4.3. Targeting Tumor Angiogenesis.

It is unclear if this section focuses on aggressive pituitary tumors and pituitary carcinomas or includes nonfunctioning pituitary tumors, which are less aggressive. I would request the authors to clarify the focus of this section. 

4. Discussion and 5. Conclusions

I agree with the authors' statement that clinical trials are needed to overcome the challenges. However, why are there no such trials? Would the authors have any suggestions for the readers? 

Reference 81. The journal name in English is "Japanese Journal of Neurosurgery". It is very strange to describe the journal name in Japanese. 

Reference 97: The style should be corrected. 

Author Response

Referee #2:  

Comment 1: The rationale for the search terms is not very clear. Why was the term "Immunotherapy" not included? This could be one of the emerging treatment options.  

Response 1: The primary focus of the manuscript was to review existing therapies that have been studied in clinically nonfunctioning pituitary tumors. The limited studies on immunotherapy use in CNFPTs have now been included (376-389). 

Comment 2: The rationale for selecting articles limited in English, Portuguese, Japanese, Spanish, and Italian is not clear. The authors could consider describing the search terms used for identifying and reviewing articles on medical therapy. The current search terms used in this project may be leading to the omission of some key articles.  

Response 2: A fuller description of the search terms used for the manuscript is now included.   

Comment 3: Reference(s) for silent somatotroph tumors (Line 91-93) are missing. It would be Reference 16.  

Response 3: The appropriate reference for this section has now been included.  

Comment 4: Reference(s) for null cell tumors (Line 112-115) are missing.  

Response 4: These additional references have now been included.  

Comment 5: References for SSTRs expression in CNFPTs (Line 163-165) are missing, likely due to the search terms used in the project. However, it seems to result in missing key articles. The authors could consider citing the following sentences, as they report SSTRs expression profiles in nonfunctioning pituitary tumors, including silent corticotroph tumors.   

Taboada GF, Luque RM, Bastos W, Guimarães RF, Marcondes JB, Chimelli LM, Fontes R, Mata PJ, Filho PN, Carvalho DP, Kineman RD, Gadelha MR. Quantitative analysis of somatostatin receptor subtype (SSTR1-5) gene expression levels in somatotropinomas and non-functioning pituitary adenomas. Eur J Endocrinol. 2007 Jan;156(1):65-74. doi: 10.1530/eje.1.02313. PMID: 17218727.  

Nielsen S, Mellemkjaer S, Rasmussen LM, Ledet T, Olsen N, Bojsen-Møller M, Astrup J, Weeke J, Jørgensen JO. Expression of somatostatin receptors on human pituitary adenomas in vivo and ex vivo. J Endocrinol Invest. 2001 Jun;24(6):430-7. doi: 10.1007/BF03351043. PMID: 11434667.  

Gabalec F, Drastikova M, Cesak T, Netuka D, Masopust V, Machac J, Marek J, Cap J, Beranek M. Dopamine 2 and somatostatin 1-5 receptors coexpression in clinically non-functioning pituitary adenomas. Physiol Res. 2015;64(3):369-77. doi: 10.33549/physiolres.932821. Epub 2014 Dec 22. PMID: 25536318.  

Neto LV, Machado Ede O, Luque RM, Taboada GF, Marcondes JB, Chimelli LM, Quintella LP,  

Niemeyer P Jr, de Carvalho DP, Kineman RD, Gadelha MR. Expression analysis of dopamine receptor subtypes in normal human pituitaries, nonfunctioning pituitary adenomas and somatotropinomas, and the association between dopamine and somatostatin receptors with clinical response to octreotide-LAR in acromegaly. J Clin Endocrinol Metab. 2009 Jun;94(6):1931-7. doi: 10.1210/jc.2008-1826. Epub 2009 Mar 17. PMID: 19293270; PMCID: PMC2730344.  

Tateno T, Kato M, Tani Y, Oyama K, Yamada S, Hirata Y. Differential expression of somatostatin and dopamine receptor subtype genes in adrenocorticotropin (ACTH)-secreting pituitary tumors and silent corticotroph adenomas. Endocr J. 2009;56(4):579-84. doi: 10.1507/endocrj.k08e-186. Epub 2009 Mar 24. PMID: 19318729.  

Response 5: These references have now been included.   

Comment 6:  Table 3 does not show a summary of the results of patients with CNFPTs treated with dopamine agonists. It does show Table 2.  

Response 6: This has been corrected, and Table 3 has now been included in the revised manuscript. 

Comment 7: The authors could consider adding the following two articles to review dopamine agonist therapy.  

Capatina C, Poiana C. Dopamine Agonists in the Management of Non-Functioning Pituitary Adenomas. Acta Endocrinol (Buchar). 2021 Jul-Sep;17(3):377-382. doi: 10.4183/aeb.2021.377. PMID: 35342478;  

Cooper O, Greenman Y. Dopamine Agonists for Pituitary Adenomas. Front Endocrinol (Lausanne). 2018 Aug 21;9:469. doi: 10.3389/fendo.2018.00469. PMID: 30186234; 

Response 7: These references have now been included.  

Comment 8: Fig. 1 suggests the FR-mediated endocytosis. However, there is no explanation in this section. Also, functions of FR in pituitary tumorigenesis are uncelar, though the section describes FR expression and its role in the activation of the Notch3 signaling pathway, leading to cell proliferation. More details would be required.  

Response 8: A fuller explanation of FR-mediated endocytosis has now been included (lines 395-402).   

Comment 9: The font size is smaller in section 4. Discussion and 5. Conclusion compared to other sections (1. Introduction, 2. Materials and Methods, 3. Results), which is strange.  

Response 9: The font size has been standardized throughout the manuscript. 

Comment 10: It is unclear if this section focuses on aggressive pituitary tumors and pituitary carcinomas or includes nonfunctioning pituitary tumors, which are less aggressive. I would request the authors to clarify the focus of this section.  

Response 10: The primary focus of the manuscript was to review outcomes to the various medical therapies that have been used to treat clinically nonfunctioning pituitary tumors. Some of these therapies were only used to treat “aggressive” clinically nonfunctioning pituitary tumors (Temozolomide) and this has been specified in the manuscript.   

Comment 11: I agree with the authors' statement that clinical trials are needed to overcome the challenges. However, why are there no such trials? Would the authors have any suggestions for the readers? 

Response 11: We believe that clinical trials have not been conducted in this field as clinically nonfunctioning tumors, unlike functioning pituitary adenomas, do not secrete a trackable biomarkers, making it challenging to assess short-term treatment efficacy. Given the often slow progression of CNFPTs, clinical trials would likely take five to ten years and large numbers of patients to document a statistically robust change in tumor size to obtain drug approval.   

Comment 12: Reference 81. The journal name in English is "Japanese Journal of Neurosurgery". It is very strange to describe the journal name in Japanese.   

Response 12: This has been corrected.    

Comment 13: Reference 97: The style should be corrected.   

Response 13: The reference style has been corrected.   

  

Round 2

Reviewer 2 Report

Comments and Suggestions for Authors

The authors addressed most of the reviewer’s questions. However, they did not provide a response regarding the rationale for selecting specific articles.
In their reply, the authors did not clarify the criteria for limiting the selection to articles written in English, Portuguese, Japanese, Spanish, and Italian. It remains unclear why these languages were chosen, while other languages, such as French, were excluded.
Furthermore, the manuscript was prepared by three authors. Are they proficient in these five languages, and was this proficiency a determining factor in selecting these five languages? Or did the authors utilize language services or applications to choose articles written in these five languages?

Very minor points: 

DR, Dopamine Receptor in the legend for Table 3 would not be needed, as there is no DR in the table. 

Table 3 in the supplemental file is still Table 2. 

Author Response

Comment 1: The authors addressed most of the reviewer’s questions. However, they did not provide a response regarding the rationale for selecting specific articles. 
In their reply, the authors did not clarify the criteria for limiting the selection to articles written in English, Portuguese, Japanese, Spanish, and Italian. It remains unclear why these languages were chosen, while other languages, such as French, were excluded. 
Furthermore, the manuscript was prepared by three authors. Are they proficient in these five languages, and was this proficiency a determining factor in selecting these five languages? Or did the authors utilize language services or applications to choose articles written in these five languages? 

Response 1: We should have clarified that our literature search included studies conducted in these regions, but we did not specifically select for articles based on these regions or languages. We wished to highlight the global scope of this review. Additionally, all studies were accessed in English. We have edited the wording of our manuscript to reflect this.     

Comment 2:  Very minor points: DR, Dopamine Receptor in the legend for Table 3 would not be needed, as there is no DR in the table.  

Response 2: We have removed DR from the legend of Table 3 

Comment 3: Table 3 in the supplemental file is still Table 2. 

Response 3: Table 3 is now included in the supplemental file.